# A Bioelectronic System to Measure the Glycolytic Metabolism of Activated CD4+ T Cells

**DOI:** 10.3390/bios9010010

**Published:** 2019-01-09

**Authors:** Suzanne M. Crowe, Spyridon Kintzios, Grigoris Kaltsas, Clovis S. Palmer

**Affiliations:** 1Life Sciences Discipline, Burnet Institute, Melbourne, VIC 3001, Australia; suzanne.crowe@burnet.edu.au; 2Department of Infectious Diseases, Monash University, Melbourne, VIC 3004, Australia; 3Infectious Diseases Department, The Alfred hospital, Melbourne, VIC 3004, Australia; 4Laboratory of Cell Technology, School of Food Science, Biotechnology and Development, Agricultural University of Athens, 11855 Athens, Greece; skin@aua.gr; 5Department of Electrical and Electronics Engineering, microSENSES lab, University of West Attika, 12244 Athens, Greece; G.Kaltsas@uniwa.gr

**Keywords:** HIV, glycolysis, Glut1, T cells, metabolism, bioelectronics, immunometabolism, mitochondria

## Abstract

The evaluation of glucose metabolic activity in immune cells is becoming an increasingly standard task in immunological research. In this study, we described a sensitive, inexpensive, and non-radioactive assay for the direct and rapid measurement of the metabolic activity of CD4+ T cells in culture. A portable, custom-built Cell Culture Metabolite Biosensor device was designed to measure the levels of acidification (a proxy for glycolysis) in cell-free CD4+ T cell culture media. In this assay, ex vivo activated CD4+ T cells were incubated in culture medium and mini electrodes were placed inside the cell free culture filtrates in 96-well plates. Using this technique, the inhibitors of glycolysis were shown to suppress acidification of the cell culture media, a response similar to that observed using a gold standard lactate assay kit. Our findings show that this innovative biosensor technology has potential for applications in metabolic research, where acquisition of sufficient cellular material for ex vivo analyses presents a substantial challenge.

## 1. Introduction

Whilst considerable research has focused on the role of antigenic signals, co-stimulation, and cytokines in guiding T cell responses, at a fundamental level, it is cellular metabolism that regulates T cell function and differentiation, and that consequently influences the final outcome of the adaptive immune response [1]. The growth, function, and survival of an activated lymphocyte depend on a pronounced increase in the glucose metabolism that is directly regulated and that has a profound impact on T cell survival and function. It is currently established that T cell metabolic reprogramming from oxidative to glycolytic metabolism affects the functions of T cells, and it impacts T cell physiology and pathological processes [2,3,4,5,6]. 

The progress in the field of immunometabolism has been hampered by the lack of techniques to measure metabolic activities in subpopulations of cells available within the limited blood samples that can be obtained with human ethics approval. Approaches to circumvent this problem include the measurement of glucose transporters, glucose uptake, and intracellular lactate production using flow cytometry [7]. Indeed, using these methods, we have identified a population of CD4+ T cells that express the glucose transporter 1 (Glut1) in humans [7]. Glut1 is the major glucose transporter on T cells [8,9] and elevated levels of Glut1 on the CD4+ T cells in HIV-infected subjects are associated with increased glycolytic activity, a metabolic profile associated with cellular activation, propensity of HIV infection, and cell loss [10,11,12,13]. 

Stimulation of Glut1 is frequently concurrent with, and/or coordinately upregulated by the activation of the Na^+^/K^+^ channels in different cell types [14]. In addition, the activation of cellular receptors usually causes transient or sustained increases in the acidification rate, i.e., the excretion of lactic and carbonic acids formed during glycolysis [15]. Therefore, the study of the ionic flux through the cell membrane can be a useful tool for the indirect determination of the glycolytic flux in T cells. Changes in cell culture acidification can be rapidly measured using ultra-sensitive electrodes with a small portable microcontroller-based electronic interface. Such devices require considerably less training and are less expensive than the more complex systems, e.g., high-end flow cytometers.

More recently, the XFe Extracellular Flux Analyzer (Seahorse Bioscience) platform of metabolic assays has revolutionized research in the field of immunometabolism [16,17,18,19,20], and of the Warberg effect that is now being intensely studied in cancer research [21]. Unfortunately, this equipment is expensive, requires specialized training, and ongoing costly consumables. 

Simpler and less expensive approaches to measure glycolytic metabolism are needed to allow cheaper access to technology and expansion of the field of immunometabolism. Emerging biosensor technologies allow for high throughput screening of changes in the extracellular microenvironment. These technologies vary from measuring the bioelectrical impedance of adherent cells to microphysiometry and the measurement of surface pH. Among these, the Bioelectric Recognition Assay (BERA) has been frequently used to measure cellular interactions with bioactive substances through the determination of electrical conductivity in the immediate vicinity of a cell cluster. Based on an advanced electronic platform and a dedicated read-out software, a BERA biosensor differentiates itself from other diagnostic methods through its superior speed (1–3 min) and sensitivity (detecting analyte concentrations less than one part in a trillion), while additional attractive features include low cost, portability, and high throughput capacity [22]. 

In the present study, we reported the development of a novel, customized Cell Culture Metabolite Biosensor device, designed on similar concepts to the BERA, to directly and rapidly measure the metabolic activity of CD4+ T cells in culture by measuring the acidification of culture media as a result of changes of the cellular glucose transport.

## 2. Methods

### 2.1. Metabolite Biosensor Device

A customized device was designed and developed to quantify the acidification in cell-free culture media, at the same time allowing for high throughput screening and high speed of assay. The device, a modification of the platform reported by Banerjee et al. [23] and Apostolou and collegues, [24] is a portable cell membrane potential monitoring and data acquisition system, having a replaceable guide bearing eight pairs of electrodes connecting on the underside (Figure 1A,B). The detection system was designed in four levels.

#### 2.1.1. The Probing Electrodes

The biomaterial housing consists of eight wells with pre-defined dimensions, where the electrodes can fit to interact with the corresponding samples. The electrodes (made of AgCl) are located on a FR4 carrier and are defined by special insulated silver wires with a diameter of 0.5–1.0 mm (Figure 1C). Ag/AgCl electrodes were prepared by immersing silver electrodes (99.99% Ag, Alfa Aesar, Karlsruhe, Germany) into 0.25 mol/L FeCl_3_ + 0.2 mol/L HCl. The AgCl electrode system used can directly measure changes in the conductivity of the extracellular microenvironment as a means of bioelectric profiling of the cell responses to various treatments. The biosensor was fully reusable and the same set of electrodes was used after rinsing with buffer or deionized water in subsequent experiments.

#### 2.1.2. The Microcontroller-Based Electronic Interface

The electronic interface was directly connected to the electrodes to acquire the evaluation data. The main part of the system was implemented by the AVR XMEGA series microcontroller running at 10 MHz. The microcontroller (μC) was equipped with a 12-bit Analog to Digital Converter capable of measuring 8 × 4 differential channels. The eight differential signals from the electrodes were directly connected to the first eight channels of the ADC for continuous monitoring. The sampling rate of the device can increase to 2 Msps, adequate for the present application. The μC also included a USART interface, to achieve direct communication to a Personal Computer.

#### 2.1.3. The Algorithm for Data Acquisition and Signal Processing

An algorithm was designed and developed for the continuous monitoring of the voltage variations in the eight wells of the biomaterial. One of the four 16-bit hardware timers was utilized to produce very accurate interrupts every 1 ms. The reading of each channel occurs every 20 ms, thus a full cycle requires 160 ms. To avoid unwanted spikes in voltage readings, a digital filter was implemented in the acquisition algorithm which rejected any reading that varied more that 80% from the average of the three previous readings. To ensure that further updates are feasible, the device can incorporate various programs that can be developed and incorporated into the system. Critical process parameters can be stored in the EEPROM of the device in order to provide the flexibility of simultaneous monitoring of several contaminants in a multi-component sample.

#### 2.1.4. The Labview Interface

A Labview program was developed for data manipulation and storage. All the data from the μC were gathered by the Labview algorithm for real time representation in a graphical interface. The data were stored in the PC for further processing. Additionally, an LCD screen was installed for enhanced portability. In both cases, the user was able to control the operation of the complete system and to visualize the real time data, as well as the processed parameters that indicated the measurement results. Details of the bioelectronics system are shown in the form of a block diagram in Figure 1D.

## 3. Preparation, Storage, and Thawing of Human Peripheral Blood Mononuclear Cells (PBMCs)

Peripheral blood mononuclear cells (PBMCs) were isolated by density gradient centrifugation (Lymphoprep, Axis Shield), using a standard approach as previously described in Reference [25]. Fresh blood samples from healthy individuals recruited in Melbourne, Australia were collected in EDTA, citrate, or heparin anticoagulant tubes. PBMC preparations began within 1 h of venepuncture. Exclusion criteria included self-reported infections, vaccination, and physical trauma prior to participation. Informed consent was obtained from blood donors and the study was approved by the ethics committee at the Burnet Institute and Alfred Hospital, Melbourne Australia. PBMCs were cryopreserved in 10% dimethyl sulfoxide (DMSO, Sigma-Aldrich) and 90% autologous plasma in 2 mL cryogenic vials (1 mL PBMCs/vial), stored for 24 h at −70 °C in a Mr. Frosty Freezing Container (Thermo Scientific), and then transferred to liquid nitrogen storage. 

Cells were thawed in a 37 °C water bath and transferred to 9 mL of supplemented RPMI-1640 medium [(10% human serum, penicillin/streptomycin (Invitrogen), 2 mM L-glutamine (Invitrogen)]. Suspensions were centrifuged to pellet cells at 300× *g* for 10 min, and then washed twice by resuspension in 9 mL RPMI-1640 medium and re-centrifugation at 300× *g* for 10 min. The cells were then resuspended in 200–1000 µL RPMI-1640 medium and viability was assessed by the Trypan Blue exclusion assay using the Countess Automated Cell Counter (Invitrogen). We typically achieved >95% viable cells and have reported that under these cryopreservation and thawing procedures, the metabolic and immunologic functionalities of the T cells were maintained [7]. 

### 3.1. Isolation and Activation of the CD4+ T Cells

The CD4+ T cells were purified from thawed PBMCs from healthy donors using the Human EasySep CD4+ T cell enrichment kit (Stem Cell, Technology Inc, Vancouver, BC, Canada). Purity (>98%) was assessed by flow cytometry after fluorescent-labeled CD4 antibody staining [7]. Purified CD4+ T cells were resuspended at a concentration of 1 × 10^6^ cells/mL in supplemented RPMI-1640 medium. Cells were stimulated with an activation cocktail consisting of PMA (100 ng/mL), ionomycin (1 ug/mL), and IL-2 (5 ng/mL) for 48 h in the absence or presence of metabolic inhibitors, and then incubated at 37 °C for 48 h with the appropriate activators in 500 µL volume in 48-well plates. For the biosensor-based analysis, only 100–200 μL of cell-free culture filtrate were required per assay.

### 3.2. Biosensor Measurements of the Cell-Free Culture Media

Following activation, the cell cultures were spun at 300× *g* for 10 min to pellet cells. Cell-free culture filtrates were frozen at −20 °C until required. For the biosensor measurement, cell-free culture filtrates were pipetted into 96-well plates and the electrodes were inserted into the wells. A heavy non-conductive object was used to keep the electrode pairs in place, allowing them to maintain contact with the culture filtrate for 3–5 min until the mV readings were stabilized before the results were recorded in duplicates (two reading channels per electrode pair). The electrodes were removed from the culture filtrates, washed thoroughly with sterilized deionized water using a uxcell 250 mL capacity squirt plastic bottle. The electrodes were then placed in 96-well plates containing deionized water to ensure the mV readings returned to baseline. Electrodes were dried by blotting gently with Kimtech Science Kimwipes before being used for subsequent culture filtrate measurements. The data were presented as delta mV, which is the difference between the baseline values and the culture filtrate readings. The device and electrodes were stored in a dry plastic custom-made storage/travelling dark plastic box to avoid exposure to varying atmospheric conditions.

### 3.3. Biosensor Measurements of Lactate Standards

The biosensor response to different concentrations of lactic acid was determined by serial dilutions of d/l-lactic acid standard (Roche) in deionized sterile water. The biosensor response was determined in 96-well plates as above.

### 3.4. Glucose Uptake Assays

#### 3.4.1. GlucMeter Reading

Glucose levels in the cell culture medium were measured using a GlucMeter, according to the manufacturer’s protocol (CESCO Bioengineering, Taichung, Taiwan) as in Reference [7]. Briefly, a disposable GlucMeter strip was placed into the GlucMeter and 2 µL of culture media was loaded onto the strip and the readings were recorded. 

#### 3.4.2. 2-NBDG Assay

The fluorescently-labeled glucose analogue, 2-*N*-(7-nitrobenz-2-oxa-1,3-diazol-4-yl) amino)-2 deoxyglucose (2-NBDG) (Invitrogen), was used to measure glucose uptake in the CD4+ T cells as previously described in Reference [7]. Briefly, cryopreserved PBMCs were thawed and recovered for 24 h at 37 °C, at 5% CO_2_ in supplemented RPMI-1640 medium. Cells were then treated with 2-NBDG 15 mmol/L 2-NBDG for 60 min, washed twice with 1× phosphate-buffered saline (PBS), and then stained with cell surface markers to identify the CD4+ T cells. The cells were then resuspended in 1× PBS and analyzed within 15 min on a FACSCalibur within the FL1 channel.

#### 3.4.3. L-Lactate Assay

Secreted L-lactate concentrations in the cell-free culture supernatants were determined using the Glycolysis L-lactate Assay Kit (Cayman Chemical) as in Reference [7]. Briefly, the cells were cultured at a concentration of 1 × 10^6^ cells/mL in supplemented RPMI-1640 medium. The cells were activated as above and treated with metabolic inhibitors and controls. Following incubation, the cell cultures were centrifuged at 300× *g* for 10 min and the supernatants were stored in 1.5 mL Eppendorf tubes at −20 °C until the L-lactate analysis. All experiments were conducted in duplicates, with three independent experiments. Absorbance readings were taken at 490 nm with a plate reader and the L-lactate concentrations of the supernatants were extrapolated based on a standard curve. 

### 3.5. Statistical Analysis

The paired T-test was used to determine the significant differences between the treatments. *p*-values < 0.05 were considered significant. All statistical analyses were performed using GraphPad Prism (version 6.0).

## 4. Results and Discussion

The biosensor system was able to detect differences in the pH value of the assayed culture media, as a measure of the differences in the conductivity of the media, as demonstrated by the results of the calibration experiments. Contrary to a conventional pH-meter, a distinct feature of the biosensor system was its ability to measure the cumulative traffic of ion charges at the working electrode over time. In other words, the biosensor could be used to assay the local ionic distribution at a given point of a solution with reference to the measuring electrode rather than the general voltage difference. An increase in the pH value was associated with a decrease in the biosensor response (expressed as changes in the mV, delta mV) (Figure 2A). We then evaluated the sensitivity and biological utility of the biosensor by evaluating the electronic response to varying concentrations of lactate and observed a dose response changes in the mV (Figure 2B).

To determine whether our bioelectronics system could be used to potentially screen metabolic modifying compounds, we challenged the CD4+ T cells with two different glucose uptake inhibitors: LY294002, a PI3K inhibitor which has previously been shown to suppress Glut1 expression and glucose uptake in CD4+ T cells, and temsirolimus, a mTORC1 inhibitor [26]. These treatments resulted in a significant reduction of glucose traffic through the cell membrane, which can be directly linked to lower glycolysis, and reduced cell membrane hyperpolarization. This in turn caused a reduction in extracellular acidification, as revealed by the reduction in the numerical biosensor response (corresponding to the increased pH of the culture medium (Figure 2C). For control, we treated purified CD4+ T cells with SB201290, an inhibitor of the p38 pathway that does not regulate glucose metabolism in CD4+ T cells [27]. 

Lactic acid is the main by-product of aerobic glycolysis in activated T cells and it is used as a measure for glucose metabolic activity. Lactic acid dissociates in water resulting in ionic lactate and H^+^. Under physiological circumstances, the pH is generally higher than the pKa, so the majority of lactic acid in the body will be dissociated and present as lactate. We accordingly reasoned that the levels of dissociated lactate would be proportional to the concentration of H^+^ ions in solution and could be quantified by our biosensor system. As shown in Figure 2C, there was a clear dose response-change in the biosensor results of the lactic dilution experiments, even though the biosensor did not selectively assay protons. The result confirms that the biosensor system has the potential to robustly detect substantial changes in the glycolytic activity in cultured cells. Other technologies also take advantage of this biological reaction for the basis of glycolytic analysis. Thus, the extracellular acidification rate (ECAR), which is principally a measure of the levels of lactic acid secreted by cells in culture medium is an accepted proxy for glycolysis, and it is the basis of the glycolysis stress test using the XFe Extracellular Flux Analyzer [28]. Importantly the analysis time using our electronic system was approximately 3 min, while the lactate assay and 2-NBDG glucose uptake took approximately 90 min and 2 h, respectively. The GlucMeter assay took approximately 10 s, but the test strips are expensive and disposable.

To compare the biosensor responses with the standardized metabolic assays, we challenged the CD4+ T cells with LY294002 and temsirolimus. As shown in Figure 3A, these drugs had no significant effects on the viability of the CD4+ T cells. Inhibition of cellular acidification as observed above, was accompanied by an almost identical reduction in the glucose uptake (Figure 3B,C) and lactate secretion by the activated CD4+ T cells (Figure 3D). Further validation of our biosensor system should include increased experimental replications, and comparisons with other metabolic readouts, such as Glut1 protein and *SLC2A1* gene (encoding Glut1) expression. 

Our novel biosensor presents considerable opportunities for the high throughput assessment of compounds interfering with the glucose metabolic activity of T cells through the reliable, rapid, and cost-efficient measurement of the cell membrane potential and extracellular acidification. Currently, this is done either by analytically studying the ion channel function with patch-clamp techniques (that suffer from low throughput and reproducibility, high cost, and complexity) [29], or fluorescence-based methods (usually requiring multiple process steps while being limited by background signals) [30]. It may also be worth mentioning that, although various biosensor systems have been developed in recent years for the determination of the toxic effects on mammalian cells, the Cell Culture Metabolite Biosensor presented in this study is the first system addressing needs in the field of immunometabolic assays in a customized fashion [23]. 

That said, our experimental approach did not aim to directly compete with the Seahorse or other commercial metabolism analyzers. Instead, it is a proof-of-concept study to investigate the feasibility of an alternative, cheaper, and faster method for measuring glucose uptake by means of a simpler, indirect measurement of the change of the cell membrane potential as an indication of the change in the glucose transport rate. Naturally, a further, more sophisticated approach would include the measurement of oxygen consumption, possibly in the framework of a future study.

In conclusion, the measurement of non-specific changes to the conductivity of the immediate cell microenvironment can be a useful tool for the assessment of the cellular response to different treatments and metabolic status, as a result of changes to the cell membrane potential and the function of membrane-based ion channels. The applicability of the novel approach of cellular bioelectric profiling has recently been demonstrated in toxicology research [31]. The present study was the first attempt to apply this emerging technology in metabolic research.

## Figures and Tables

**Figure 1 biosensors-09-00010-f001:**
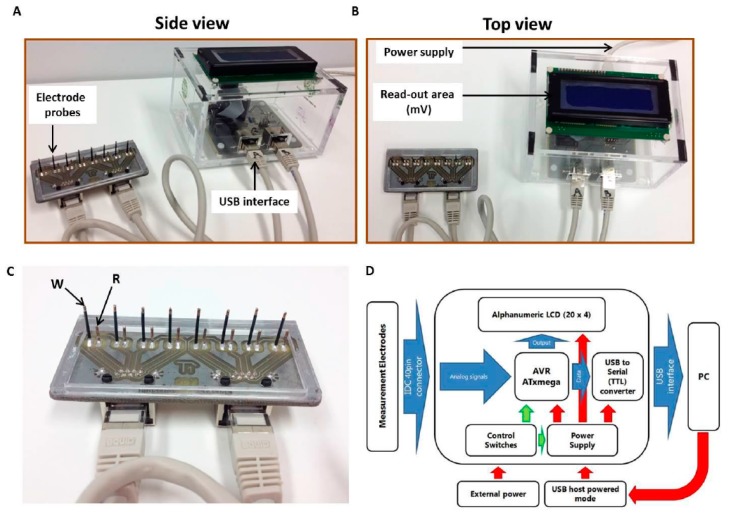
Components of the Metabolite Biosensor Device used to measure the levels of cell membrane potential and the conductivity of cell cultures. (**A**) Side view and (**B**) Top view showing the key features of the Metabolite Biosensor Device. The portable potentiometer (with the read-out area) is connected to the electrode probe through a customized interface. Using this configuration, a maximum number of eight tests can be run simultaneously, corresponding to a row of eight microwells. (**C**) Rear view of the customized electrode probe bearing eight pairs of electrodes for simultaneous signal monitoring (W = working electrode, R = reference electrode). (**D**) System block diagram of the bioelectronics assay system.

**Figure 2 biosensors-09-00010-f002:**
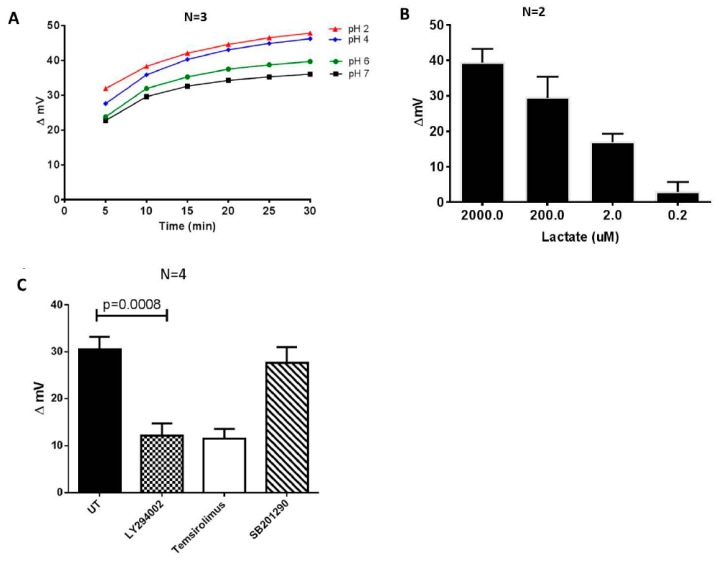
Inhibitors of PI3K (LY294002) and mTORC1 (temsirolimus) suppress the acidification and conductivity of culture media of the activated CD4+ T cells. Different pH buffers were initially used to evaluate the performance of the system. (**A**) Changes in the delta mV of different pH reference buffers over a period of 30 min. (**B**) The biosensor response to different concentrations of lactic acid. (**C**) CD4+ T cells were purified by negative selection from HIV-healthy donors and stimulated with an activation cocktail consisting of PMA (100 ng/mL), ionomycin (1 ug/mL), and IL-2 (5 ng/mL) for 48 h, replaced with fresh media, and were left untreated (UT) (0.1% carrier DMSO), or treated with either LY294002 (15 uM), temsirolimus (100 nM), or SB201290 (10 uM, p38 inhibitor) for an additional 48 h. Glycolytic activity in the culture medium was measured using the Metabolite Biosensor Device. Cells were cultured at a concentration of 1 × 10^6^ cells/mL in supplemented RPMI-1640 medium. The paired T-test was used to determine the significant differences between the treatments.

**Figure 3 biosensors-09-00010-f003:**
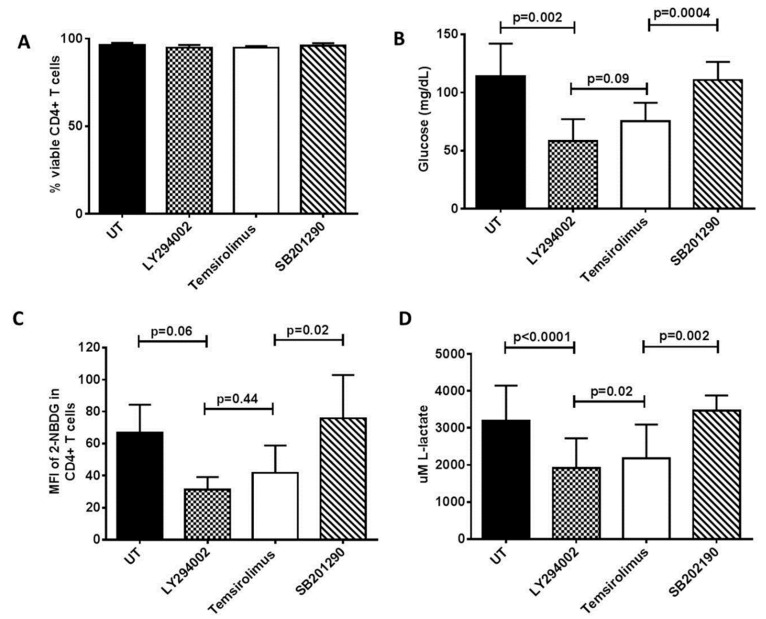
Inhibitors of PI3K (LY294002) and mTORC1 (temsirolimus) suppress glucose uptake and lactate production by activated CD4+ T cells. CD4+ T cells were purified by negative selection from HIV-healthy donors, stimulated (as described in the legend of Figure 2) with PMA, ionomycin, and IL-2 (5 ng/mL) for 48 h, replaced with fresh media, and then left untreated (UT) or treated with either LY294002 (15 uM), temsirolimus (100 nM), or SB201290 (10 uM, p38 inhibitor). (**A**) Viability of the CD4+ T cells as measured by trypan blue exclusion assay. (**B**) Data showing the amount of glucose taken up by the CD4+ T cells in culture using the GlucMeter. (**C**) 2-NBDG uptake by the CD4+ T cells as measured using flow cytometry. (**D**) L-lactate levels in CD4+ T cell culture medium, measured using an L-lactate assay kit as described in the Section 2. All experiments were conducted in duplicates, with three independent experiments. The paired T-test was used to determine significant treatments between the treatments.

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
