# Peer review of "A Bioelectronic System to Measure the Glycolytic Metabolism of Activated CD4+ T Cells"

_biosensors, 2019, doi:10.3390/bios9010010_

Reviewer 1 Report

I have read this article thoroughly, which describes the development of a multi electrode device for measurement of metabolic activity in plate style experiment.

I tried to find fault with the manuscript but have to say I found it to be an enjoyable read and a sound paper and do not feel it requires any modification or revision.

The presented data include controls (e.g. measurement of different lactic acid solutions and pH values) and are therefore convincing of the ability to measure metabolic activity in situ.  I can foresee quite a lot of applications for a system such as this one.

Author Response

We thank the reviewer for his comments

Reviewer 2 Report

The authors present an electrochemical biosensor to analyse the metabolic activity of cells based on changes in pH due to acidification of the culture medium by lactate. The authors show that the echem setup is capable of indirect monitoring of changes in cell metabolism. The comparison with more specific bioassays (fluorescent-glucose, glucose-meter as well as standard colorimetric plate read-out) show that their approach measures similar T cell responses. 

 Comments:

 1) Methods: It is hard to follow the details of the experimental setup when only references to other publications and manufacturers manuals are provided. The manuscript lacks methods, volumetric measures as well as concentrations of liquids so that anybody can reproduce these experiments

 2) Ref 24: The presented system is based on REF24, however, there is no indication of what electrode materials were used in the current study (see Materials)?

 3) The authors present that they can probe glucose uptake/metabolism and claim that their biosensor is similar but cheaper than the Seahorse system. To prove this claim, the authors should also present how they can measure oxygen consumption in parallel as an important control parameter to be tested during aerobic glycolysis.

 4) Line 164-66: How can the cumulative unspecific traffic of ion charges at the WE over time be a feature of the presented biosensor, even though less specific and selective than other biosensors?

 5) The curves of Figure 2A and 2B seem to be the same and therefore redundant. Is there any reason for this second graph?

6) Figure 2C: It would be interesting to see an additional plot that shows lactic acid conc vs. actual pH, as well as more repetitions on the BERA measurements (n=2 is to few).

7) Figure 2CD: If their sensor is linear to pH as well as lactic acid as claimed, why dont the authors plot pH or lactate conc. instead of mV?

8) Can the authors provide a PCR analysis of GLUT-1 for the primary T Cells

9) Is the biosensor reusable? The authors should show reusability data for lactic acid measurements in supplemented medium?

10) The authors should compare the assay time for 12 samples (UT and 3 compounds, n=3) of the BERA compared to the other 3 assays to understand the assay speed. Are there big differences in the timeframe?

 Author Response

The authors present an electrochemical biosensor to analyse the metabolic activity of cells based on changes in pH due to acidification of the culture medium by lactate. The authors show that the echem setup is capable of indirect monitoring of changes in cell metabolism. The comparison with more specific bioassays (fluorescent-glucose, glucose-meter as well as standard colorimetric plate read-out) show that their approach measures similar T cell responses.

1)      Methods: It is hard to follow the details of the experimental setup when only references to other publications and manufacturers manuals are provided. The manuscript lacks methods, volumetric measures as well as concentrations of liquids so that anybody can reproduce these experiments

Response: The manufacturing and set-up of the biosensor system is described in detail and in such way that an expert in electronic engineering can readily reproduce the device used in our experiments. We have now provided additional details in the Materials and Methods sections.

2) Ref 24: The presented system is based on REF24, however, there is no indication of what electrode materials were used in the current study

Response: The electrodes were made of silver chloride (AgCl). This detail has been included in the Materials & Methods section of the revised manuscript lines 137-142.

2)      The authors present that they can probe glucose uptake/metabolism and claim that their biosensor is similar but cheaper than the Seahorse system. To prove this claim, the authors should also present how they can measure oxygen consumption in parallel as an important control parameter to be tested during aerobic glycolysis.

Response: Our experimental approach does not aim to directly compete with the Seahorse or other commercial metabolism analysers; instead, it is a proof-of-concept study to investigate the feasibility of an alternative, cheaper and faster method for measuring glucose uptake by means of a simpler, indirect measurement of the change of the cell membrane potential as an indication of the change in glucose transport rate. Naturally, a further, more sophisticated approach would include the measurement of oxygen consumption, possibly in the framework of a future study.

This definition has been added to the Conclusion of the revised manuscript.

3)      Line 164-66: How can the cumulative unspecific traffic of ion charges at the WE over time be a feature of the presented biosensor, even though less specific and selective than other biosensors?

Response: This is due to the fact that the biosensor measures the change of the conductivity of the immediate cell microenvironment, which itself is associated with changes of the cell membrane potential and the function of membrane-based ion channels. Though said changes are not specific themselves, they still reflect the cellular response to different treatments and metabolic status (cellular bioelectric profiling). This definition has been added to the last paragraph of the Conclusion of the revised manuscript, in form of the following statement:

“In conclusion, the measurement of non-specific changes of the conductivity of the immediate cell microenvironment can be a useful tool for the assessment of the cellular response to different treatments and metabolic status, as a result of changes of the cell membrane potential and the function of membrane-based ion channels. The applicability of the novel approach of cellular bioelectric profiling has been recently demonstrated in toxicology research [31]. The present study is the first attempt to apply this emerging technology in metabolic research.”

A new reference [31] was added.

 4)      The curves of Figure 2A and 2B seem to be the same and therefore redundant. Is there any reason for this second graph?

Response: Figure 2B was meant for direct comparison between two pH values but we took the reviewer’s point and removed Figure 2B from the revised manuscript.

 5)      Figure 2C: It would be interesting to see an additional plot that shows lactic acid conc vs. actual pH, as well as more repetitions on the BERA measurements (n=2 is to few).

Response: We thank the reviewer for his suggestions but we are unable to conduct additional experiments to meet to the revision deadline and due to absence of consumables. Furthermore there is a consistent trend between mV and the different lactate concentrations, suggesting a direct relationship between these two variables. We have included the future need for increased replications in the discussion of the revised manuscript.

6)      Figure 2CD: If their sensor is linear to pH as well as lactic acid as claimed, why don’t the authors plot pH or lactate conc. instead of mV?

Response: We have plotted lactate concentrations in Figure 3D which reflect similar trends as changes in mV (updated Fig 2C).

7)      Can the authors provide a PCR analysis of GLUT-1 for the primary T Cells

Response: We thank the reviewer for this suggestion but this is beyond the scope of the study, which is to test the feasibility of our electronic system to study surrogate of metabolic activity. However we have included this suggestion in our discussion as additional approach to validate  our system. Lines 264-266.

9) Is the biosensor reusable? The authors should show reusability data for lactic acid measurements in supplemented medium?

Response: Yes, the biosensor is fully reusable and the same set of electrodes was used after rinsing with buffer in subsequent experiments.  This definition was added in the Materials & Methods section of the revised manuscript. 

8)      The authors should compare the assay time for 12 samples (UT and 3 compounds, n=3) of the BERA compared to the other 3 assays to understand the assay speed. Are there big differences in the timeframe?

Response: The BERA assay is 3 minutes. The timeframe for the GlucMeter assay is approximately 10 seconds but test strips are disposable and expensive. Glucose uptake assay took approximately 2h and lactate assay approximately 90 minutes. We have added this information to the results and discussion lines 252-255.

Reviewer 3 Report

1.     Please clarify the mechanism of pH test. Which kind of electrode was used? Is the electrode responsive to proton only?

2.     Is there buffer in the cell culture? Does the buffer capacity matter for the pH-based test?

3.     On page 8: It should be noted, however, that comparisons of conditions where subtle differences in metabolic activity are expected might not be appropriate for this version of our bioelectronics system. Could you further explain this?

4.     In the conclusion section, patch clamp was mentioned as a technology for comparison. Is patch clamp able to test pH change?

Author Response

1.        Please clarify the mechanism of pH test. Which kind of electrode was used? Is the electrode responsive to proton only?

Response: The AgCl electrode system used is actually measuring changes in the conductivity of the extracellular microenvironment, as a means of bioelectric profiling of cell responses to various treatments. It does not measure directly the pH value of the medium. This was defined as the principle of the BERA method in the Introduction section. It was also added as a definition in the Materials & Methods section of the revised manuscript.

Furthermore, in the Results & Discussion section (first paragraph) of the revised manuscript the following statement was added:

“The biosensor system was able to detect differences in the pH value of the assayed culture media, as a measure of differences in the conductivity of the media...”

In the same sense, the following statement was added in the third paragraph of the Results & Discussion section of the revised manuscript:

“As shown in Fig. 2C, there was a clear dose response-change in the biosensor results of the lactic dilution experiments, even though the biosensor does not selectively assay protons.”

 2.        Is there buffer in the cell culture? Does the buffer capacity matter for the pH-based test?

Response: We used standard RPMI cell culture solution with low buffering capacity

3.        On page 8: It should be noted, however, that comparisons of conditions where subtle differences in metabolic activity are expected might not be appropriate for this version of our bioelectronics system. Could you further explain this?

Response: This statement was omitted from the revised manuscript.

4.     In the conclusion section, patch clamp was mentioned as a technology for comparison. Is patch clamp able to test pH change?

Response: Patch clamp was explicit mentioned as a comparative technology for measuring changes in cell membrane potential and ion traffic, not pH change.

Round  2

Reviewer 2 Report

The authors corrected their manuscript well, however their are still some minor things that need to be addressed prior to publication:

1) Ad AgCl:

How were the silver electrodes produced? Are those commercial AgCl electrodes or chemically modified silver wires? How ere the wires chlorinated?

2)Ad Experimentals:

 The remark that an electrical engineer could reproduce this device (which is obvious) is appreciated, however, the experimentals still lack necessary volumetric details and proper protocol steps for the bioassay aspects to be reproducible (L-lactate). The statement that a commercial assay was used without the actual experimental procedure (at least as a brief summary) should be avoided.

Similarly, the T cell isolation from the healthy donor PBMC samples should at least be sketched? Were PBMCs bought commercially at a cell bank or isolated from whole blood or apheresis collection? How was the assay used?

Author Response

The authors corrected their manuscript well, however their  are still some minor things that need to be addressed prior to publication:

 1) Ad AgCl: How were the silver electrodes produced? Are those commercial AgCl electrodes or chemically modified silver wires? How ere the wires chlorinated?

Response: The silver electrodes (99.99% Ag, Alfa Aesar, Karlsruhe, Germany) were covered by a stable AgCl film by immersing them into 0.25 mol/l FeCl3 + 0.2 mol/l HCl. This is now indicated in the methods.

2)Ad Experimentals: The remark that an electrical engineer could reproduce this device (which is obvious) is appreciated, however, the experimentals still lack necessary volumetric details and proper protocol steps for the bioassay aspects to be reproducible (L-lactate).

The statement that a commercial assay was used without the actual experimental procedure (at least as a brief summary) should be avoided.

Response: We have provided more details on the Bioelectronic measurement and have  modified the sequence of  methodologies to allow better flow. We have also added additional details of the L-lactate assay. Please note we have also provided more detailed experimental information in the figure legend

 Similarly, the T cell isolation from the healthy donor PBMC samples should at least be sketched? Were PBMCs bought commercially at a cell bank or isolated from whole blood or apheresis collection? How was the assay used?

Response: The PBMC preparation is a very standardized method therefore does not warrant a sketch. However we appreciate that the readership of this journal might not be familiar with the process so we have given a more detailed account of the experimental procedure.